# Ball Screens in the Men’s 2019 Basketball World Cup

**DOI:** 10.3390/ijerph20010059

**Published:** 2022-12-21

**Authors:** Iván Prieto-Lage, Christopher Vázquez-Estévez, Adrián Paramés-González, Juan Carlos Argibay-González, Xoana Reguera-López-de-la-Osa, Alfonso Gutiérrez-Santiago

**Affiliations:** 1Observational Research Group, Universidade de Vigo, 36005 Pontevedra, Spain; 2Education, Physical Activity and Health Research Group (Gies10-DE3), Galicia Sur Health Research, Institute (IIS Galicia Sur), SERGAS-UVIGO, 36208 Vigo, Spain

**Keywords:** performance, coaching, critical moments, t-patterns, observational study

## Abstract

Background: The objective of this research was to analyse the effectiveness and search for successful patterns in ball screens in the men’s 2019 Basketball World Cup. Methods: The sample consisted of 515 ball screens obtained in critical moments. LINCE software was used as a registration instrument by means of an observational instrument designed ad hoc. A descriptive analysis and chi-square tests (χ2) were performed with SPSS 25.0 and a T-patterns analysis with Theme 5 software. Results: The results indicate that the criteria that have the most influence on this type of action are the result of the team executing the screen (winning, losing or tying), the type of offense and the defence used on the ball screen. The most representative patterns of success tend to take place between 9–16 s of possession, with the screeners being inside players and the screened players being outside players, and it is performed in the upper areas of the court against an individual type of defence and ending with the screened player advancing towards the basket or passing to an open teammate. Conclusions: The data obtained will enable the coaching staff to train ball screens in accordance with specific game situations.

## 1. Introduction

Currently, basketball is one of the most statistically analysed sports, which allows coaches to evaluate the effectiveness of technical-tactical aspects during the course of a game or over the course of a season [1]. Because of this, it is more difficult to obtain offensive advantages, and consequently, scoring becomes more difficult [2]. Identifying these offensive patterns of play will become a determining factor in improving the tactical performance and decision-making present during a match, as it will generate the best opportunities to obtain open shots with the least opposition [3,4,5]. These situations that cause a mismatch in the defence and, consequently, the creation of unopposed space, are known as space creation dynamics [6]. Along the same line, other authors [7] consider that offensive strategies aim to achieve optimal shooting situations, understanding them as those executed by the best shooters in a context where the highest percentages of success are ensured. Collective actions will increase uncertainty, giving it a certain degree of complexity and preventing the opponent from reacting effectively, as opposed to if the actions were carried out individually [8].

The minimum expression of collective play is ball screen, also known as pick-and-roll or on-ball screen [9]. Ball screen has been defined as the legal interposition of an offensive player towards a defender with the aim of freeing a teammate to shoot or receive a pass [10]. Similarly, other researchers [11] define ball screen as the basic technical–tactical element of cooperation between two players possessing the ball, where the screener performs a screen or “pick” with the aim of the screened player getting a positive situation and advantage over his defender and, after this, the screener advances (roll) towards the basket to receive a pass [12,13]. The advantages of using this technical–tactical element lie in helping the dribbler to advance towards the basket, getting free throws or causing changes of assignment in the opposing defence [7]. For this, the players involved must take into account the place on the court where it is performed (near or far from the basket, centrally or laterally), the position of the other players not involved in the action or the remaining time of possession [9]. Given that ball screen is the most used collective element during the offensive phase [5,12,14,15], several scientific studies have focused on its study. One of the first studies on this subject [12], showed that ball screens were one of the most important technical-tactical elements in the final actions, accounting for 12.7% of collective actions and representing an average of 1.08 points per possession. In another study [6] it was found that ball screens were used in 34.8% of the offensive actions in the 2002 men’s Basketball World Cup in Indianapolis (USA). Similar results were found in another study with a use of 50% during the first 5 min, 40.6% during the next 30 min and 36.7% in the last 5 min in the men’s ACB league in Spain [16]. These same authors pointed out that the effectiveness of possessions varies according to gender, game period and technical–tactical indicators. Other authors [11] pointed out that the effectiveness of screens is greater when they are made in transition, in the last 8 s of possession and in the upper areas of the court. Given the importance of this technical–tactical element in offense situations, the way in which they are defended is also a determining factor in the final result of possession, which has motivated several investigations [2,13,17,18,19].

In elite basketball, the last possessions determine the final outcome of the game [3]. The last five minutes and overtime are defined as the critical moments [20]. In addition to the remaining time, the difference in the score (the psychological barrier stands at 10 points) also seems to be a determining factor [21,22]. Considering this series of conditioning factors, and adding that the level of the opponent or playing as a home or away team will affect the final score [16,23], it is of interest to study ball screens in situations that meet these requirements. A good scenario could be the Basketball World Cups, where the best teams and players in the world compete against each other, providing the opportunity to study and analyse technical–tactical behaviours [24].

Taking into account the influence of successful ball screen actions on the final result of the match and considering that a high number of matches are settled in the last few minutes, it seems advisable to carry out studies that combine the analysis of both variables. In this way, coaches will be able to obtain valuable information for the preparation of their teams offensive and defensive tactics. Therefore, the main objective of this research is to analyse the effectiveness and successful patterns in the last five minutes of the last quarters and overtime of games with a final score equal to or lower than 10 points of the 2019 men’s Basketball World Cup.

## 2. Materials and Methods

### 2.1. Design

Observational study [25] was used to analyse ball screen actions in critical moments of basketball. This type of methodology is the most appropriate for the study of sports, as it can be analysed in its usual context and dynamics [26], and as it has been tested in basketball through various publications [2,5,7,27,28].

The observational design [29] is nomothetic, because the screens of all participating teams were studied, follow-up, because 32 games were studied throughout the 2019 Basketball World Cup and unidimensional, as there was only one level of response in the data registration.

### 2.2. Sample

The sample consisted of all ball screens in the last five minutes of the last quarters and overtime from the preliminary round, second round and final round of the 2019 men’s Basketball World Cup with a final score difference of 10 points or less (*n* = 515). As this was an observational study in a natural environment and did not involve experimentation of any kind, it was not necessary to obtain the informed consent of the competitors [30]. The study was approved by the Ethics Committee of the Faculty of Education and Sport Science (University of Vigo, Application 06-280722).

A valid ball screen was considered as one that occurred in the offensive zone and in which there was a direct interaction between at least one offensive player and one defensive player. Pick-and-slips, hand-offs and rejects were excluded. To classify ball screens into “successful” and “unsuccessful” in the criterion “outcome”, the proposal of another similar study was used [16]. Thus, a ball screen was considered successful when the screening team scored a 2- or 3-point field goal after the screened player advanced, passed to an open teammate, or when the screened or the screener player suffered a foul immediately after the screen.

A screen was considered unsuccessful when the offensive players missed a 2- or 3-point field-goal, received a block shot, committed a foul, passed to a teammate who did not shoot, lost possession of the ball or committed any violation of the rules. In addition (in order to observe the real effectiveness of ball screens), it was decided that in actions classified as successful, at most, the two players directly involved in the screen (screened and screener) should participate. A third player could participate if they did not bounce the ball once they received the ball in a shooting position. In cases where two consecutive screens occurred, if they were made by the same players, in the same area and at the same interval of possession, it was considered as a single screen. If any of the above conditions were not met, the two screens were analysed separately, and the result of the action of the first screen was “don’t shot”.

### 2.3. Instruments

The observational instrument was created ad hoc and consisted of eleven criteria and forty descriptors (Table 1 and Figure 1), which comprise a system of categories that complies with conditions of exhaustiveness and mutual exclusivity. The criteria were extracted from various research studies in the scientific literature [5,7,11,15,31]. The software LINCE v.1.4 and LINCE PLUS [32,33] were used as data registration tools.

### 2.4. Procedure

First, the videos analysed in this study were downloaded from the International Basketball Federation (FIBA) YouTube channel. Subsequently, the last five minutes of the last quarter and overtime were cut, so that a single file was created with all the games ordered by date of play. The software Filmora (v.10.1.20.15) was used to create and edit this file.

After proper training in the use of the instruments, ball screens were observed and registered by expert observers. To ensure rigour in the registration process [34], the quality of the registration data was controlled by calculating intra- and inter-observer agreement using the Kappa coefficient [35], calculated using LINCE software. Both concordances were performed with screens who did not belong to the final sample (*n* = 100; 1/5 of the sample).

After the registration of all actions, an Excel file with the sequence of behaviours was obtained. The versatility of this file allowed successive transformations to be made for the different analyses carried out during the research [36].

### 2.5. Reliability

For intra-observer agreement, a kappa value of 0.99 was obtained for Observer1 and 0.97 for Observer2. For inter-observer agreement, a kappa value of 0.97 was obtained. In cases where differences in registration were identified, the observers reached consensus before performing the full analysis. Once this test had finalized, the data on the final sample was registered.

### 2.6. Data Analysis

All statistical analyses were performed using the IBM Statistical Package for the Social Sciences, version 25.0 (IBM-SPSS Inc., Chicago, IL, USA). Statistical significance was assumed for *p* ≤ 0.05.

A descriptive analysis of the categories studied was conducted globally. Test χ2 was used to contrast the existing differences between the categories of each criterion (intra-criteria analysis), as well as to compare the existing differences between the different criteria analysed and the criteria team, scoring, possession, screening defence type and effectiveness (inter-criteria analysis).

To determine the exact sequentiality of ball screens in basketball, a T-pattern analysis was performed with Theme v.5.0 [37]. T-pattern detection is a technique that recognizes recurrent patterns such as behaviour events over time by capturing variability in timing and defines occurrences of patterns based on statistical probabilities [38]. The following search criteria were applied: (a) presence of at least 4 given T-patterns, the minimum possible that could be used and that would not generate errors in the statistical software due to excessive data; (b) redundancy reduction setting of 90% for occurrences of similar T-patterns and, (c) significance level of 0.005.

## 3. Results

Table 2 presents the descriptive analysis of the study and the χ2 intra-criteria test.

Significant differences were evident in all the criteria studied. In general, ball screens were more common in teams that were losing (46.80%), with a difference in the score of less than 4 points (52.62%), through a set play (70.7%), in the middle phase of possession (when there were between 9 and 16 s left; 56.1%), in the top areas of the court, both right and left, (55.53% and 40.39%, respectively), by inside players (87.38%) to outside players (99.22%). Most of them were in man-to-man defences (98.83%). Almost half of the screens (46.02%) were defended through switch defence, followed by over the screen (24.7%) and under the screen defence (17.48%). 

Only 29.71% of the screens were considered effective, with the most recurrent successful action being the advance to the basket of the screened player (13.75%); followed by the pass to the open teammate (7.71%). Among the unsuccessful screens, the most frequently executed action was not shooting at the basket (32.62%), followed by a missed field goal (26.02%).

On the other hand, an inter-criteria analysis was carried out between the effectiveness criterion and the rest of the criteria (Table 3). The results showed significant differences for the criteria team, type of offense and screening defence type.

In order to carry out a more specific analysis of the successful sequences of the ball screens, T-patterns have been calculated with a minimum occurrence of 4 with the Theme 5 software. A total of 1232 successful patterns were found. Table 4 shows the T-patterns where the team that executed the ball screen was winning, taking into account the difference in the score and the time of possession available.

The most representative pattern in the winning category (seven occurrences) occurred when the difference in the score was between 0 and 3 points, the offense was a set play and took place between 9 and 16 s of possession. As for the players involved, the screen was made by an inside player to an outside player. The opposing team defended the play with man-to-man defence and the screening defence type was over the screen.

Table 5 shows the patterns of effectiveness while the team was losing or drawing, taking into account the difference in the score and the remaining time of possession.

The most representative pattern in the losing category (eight occurrences) was when the difference in the score was between 0 and 3 points and with a set play. The players involved were the inside player (screener) and the outside player (screened). The opposing team was set up in individual defence and the ball screen took place with a switch. Finally, in order for the screen to be successful, the action of the screened player was to advance to the basket. The other pattern with the highest occurrence took place with the same score and type of offense. The screening area occurred on the top-left side of the court and the players involved in the action were the inside player as the screener and the outside player as the screened player. The collective defence type was individual and in the ball screen there was a switch. 

In ball screens in a tie situation, the two most representative patterns with the highest occurrence (five) were with a set play. Both occurred with between 9 and 16 s of possession remaining. As for the players involved, the screener was an inside player and the screened player was an outside player. The collective defence was man-to-man. In one of them, the screening area was located on the top-right side and in another, the action performed by the screened player was to pass the ball to the open teammate.

## 4. Discussion

The objective of this research was to analyse the effectiveness and successful patterns of ball screens in the last five minutes of the last quarters and overtime (critical moments), in games with a final score equal to or less than 10 points of the 2019 men’s Basketball World Cup.

Collective actions seem to be more effective than those involving only one player in the final minutes of a game given their ability to improve the spatial dynamics of play [3]. Considering ball screens as the most used collective element during the offensive phase [5,12,14,15], this research was postulated as an interesting object of study to contribute resources to the scientific literature of basketball.

Firstly, we observed the existence of interactions between the effectiveness of ball screens and the result of the screening team, the type of offense and the defence of the screen, reiterating the idea that group behaviour depends on temporal and contextual factors [12]. With respect to the result of the team that carried out the screen, it was observed that the teams that were losing were the ones that executed this technical–tactical action the most times (241) and with a higher effectiveness (36.1%). This data coincides with a recent study [9], which showed that teams with a score between −10 and 0 points were more effective than teams that were winning. Identical results were found in similar research [11]. One of the causes may be that these teams play more aggressively due to the score, aiming to generate defensive mismatches. The theory that teams play more aggressively when the score is close is reinforced in another study [39] where the indicators that most discriminate between winning and losing teams in this kind of games are steals and the execution of personal fouls. Similarly, other research [5] observed that teams that were losing performed more pick-and-roll actions than those that were winning (52.2% vs. 49.9%).

Regarding the type of offense, it was noted that both the highest number of positive actions (122), as well as the effectiveness (33.5%), occurred when ball screens were executed within a set play. The results obtained do not coincide with those of other similar research [11], where they showed a greater effectiveness in transition plays compared to set plays (52.9% vs. 47.1%). On the other hand, there is agreement that there is a higher success/non-success ratio in ball screens made during set plays (33.5–66.5%) compared to transition or fast-breaks offense (19.9–80.1% and 26.7–73.3%, respectively). In relation to the percentage of set plays (70.7% of the total plays), these results were similar to those reported in another research (78.5%) in the male category [40]. The higher rhythm of play obtained in this study could be related to the context of the research itself, which would be the critical moments of the match and with close difference in the score. Therefore, the final outcome of the matches is unpredictable and teams use transition play to obtain favourable scoring opportunities more quickly [21,41].

With regard to screen defence, the switch was the main way of defending the screen (46%), probably because it is the quickest way to defend a ball screen [42]. The results of this study coincide with those found in the literature [42], where it is indicated that this defence is the most used against ball screens, and in line with another study [13], where it was found that this type of defence is the second most used (23.1%), only surpassed by the over defence (40.8%). In terms of effectiveness, their results coincide with those obtained in this research (70% of unsuccessful actions), classifying it as one of the best options for defending pick-and-rolls. The problem with this defence lies in the generation of mismatches, in which case, the shorter the response time (1–4 s) the more successful the attack will be [17].

The over defence (the second most used in this study) was the most used in two other studies [13,15]. It was the least effective of the three most used, as it achieved 33.3% of effective screens compared to 15.6% with the under defence and 30% with the switch, these data coincide with a recent study [13]. The under defence was the most effective defence for defending the screen, as in 84.4% of the occasions the completion of the screen was unsuccessful. This coincides with the results of another study [43], where this defence was the most efficient in defending the ball screen (76.9%). However, the risk of this defence lies in the shot possibility provided to the screened player. Therefore, if it is performed close to the basket, it seems to be more effective to perform some defensive help or a trap instead of this type of defence [44].

In terms of pattern analysis, we can see that the majority of successful actions occur in the middle phase of possession, i.e., when there are between 9 and 16 s left, regardless of whether the teams were winning, losing or drawing and regardless of the difference in the score. However, the highest effectiveness of the screens was at the end of the possession, i.e., when there were between 0 and 8 s left (36.3%). Both findings are consistent with previous available literature [5,11,15], and could possibly be explained by the fact that at the end of possessions there is greater defensive disorganisation and greater player fatigue [45]. When teams were losing, regardless of the score difference, there is a greater number of successful actions at the beginning of possession (24–17 s) compared to the end of possession (0–8 s). This may be due to the fact that teams that are losing trying to apply fast-breaks or transitions, as these actions are characterised by their speed and effectiveness in scoring [4].

In all the most representative successful patterns, the screened player was an outside player, the screener was an inside player and the opposing defence was man-to-man; coinciding with the data obtained through polar coordinates in a recent study [2]. This may be because the inside players tend to be the tallest and heaviest, while the outside players tend to be the quickest and most agile [46]. The predominant areas for the execution of the screens, regardless of success or failure, were the upper areas of the court, both right and left (55.5% and 40.4%, respectively), corroborating the data present in the literature [2,5,11,31]. The greater use of the top-right zone may be due to the predominance of players being right-handed and thus feeling more comfortable developing their game from this position [9,47]. The most common way to end a successful ball screen is for the screened player to advance to the basket and score or for the screened player to assist a “open” teammate. This coincides with research with similar objectives [15], where it was observed that 26.1% of successful screens ended with a layup by the dribbler player and 12.9% with a pass to an open teammate. In the same line [13], they corroborated how the majority of pick-and-rolls (270) concluded with the advancement of the screened player, differentiating between 152 successful and 118 unsuccessful actions. In another study [4], it was found that the most effective way of finishing two-point baskets in the EuroLeague was after a ball screen. Likewise, it has been shown that 43% of ball screens ended with a shot by the screened player [48], but highlighting the pass to the roller or screener as the most effective option (59.74%). Recently it was shown that the most successful actions after a pick-and-roll are the shot by the screened player (1.14 points per action), the jump-shot by the screener (1.24 points) and the pass to a third player not involved in the action (1.09 points) [18].

### 4.1. Practical Implications

The study reveals the importance of recognising the patterns originating in offensive actions in which ball screen is used to obtain points in the critical phase of the match. This will allow coaches to have a more exhaustive knowledge of this technical–tactical element, being able to work on these aspects during training sessions.

From an offensive perspective, it is recommended to execute ball screens in the upper part of the court (see Figure 1) while executing a set play. The ball screens were more effective if there were less than eight seconds of possession left and the screened player was moving towards the basket.

From a defensive perspective, man-to-man defence is recommended, followed by defensive switching after the ball screen. In the same way, it is recommended that ball screens are performed in the lower areas of the court at the start of the possession (9–24 s remaining), preventing the player who was screened from advancing towards the basket.

### 4.2. Limitations of the Research and Future Perspectives

The sample is relatively small, as only one complete world championship was analysed, so practical applications should be taken with caution. The context of the object of study is very specific, so there is no research with which to compare it directly, having to use those that analyse the matches without considering the “critical situation”. It should be taken into account that, due to the type of study design established, an unsuccessful ball screen has been considered when after the execution of a ball screen there is a failure in a two or three-point shot. This should be considered in the interpretation of the research results.

As for future lines of research, it would be interesting to analyse screening defence from the point of view of the screener and the screened individually, not only jointly. On the other hand, it would be interesting to analyse successful patterns in more diverse samples such as the NBA, EuroLeague or women’s basketball competitions.

## 5. Conclusions

The data from the study indicates that teams that are losing make a greater number of ball screens and with greater effectiveness. Set plays are more effective than in transition or fast-break. Switch defence is the most used to defend this technical–tactical action.

In terms of spatial–temporal aspects, most screens are usually executed during the middle phase of possession (9–16 s), although the greatest effectiveness is at the end of possession (0–8 s). The execution zones are usually located at the top of the court, both right and left.

The screeners are usually inside players, while the beneficiaries are outside players. The collective defence of the team on which the action is executed is man-to-man. The most common way of finishing a successful ball screen is the advancement of the screened player towards the basket or the pass to an open teammate.

## Figures and Tables

**Figure 1 ijerph-20-00059-f001:**
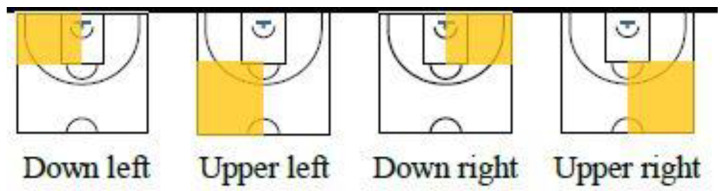
Screening Area.

**Table 1 ijerph-20-00059-t001:** Observational instrument.

Criteria	Description
Team	The screening team is winning.
	The screening team is tying.
	The screening team is losing.
Scoreboard	The difference in the scoreboard is between 0 and 3 points.
	The difference in the scoreboard is between 4 and 6 points.
	The difference in the scoreboard is between 7 and 9 points.
	The difference in the scoreboard is higher than 10 points.
Type of offense	The action occurs in a fastbreak action.
	The action occurs in a transition action.
	The action occurs in a set play action.
Possession	The action is performed with 0–8 s of possession remaining.
	The action is performed with 9–16 s of possession remaining.
	The action is performed with 17–24 s of possession remaining.
Screening Area	Top right-hand side of the court. Bottom right-hand side of the court. Top left-hand side of the court. Bottom left-hand side of the court.
Screener	The player who makes the ball screen is an inside player.
	The player who makes the ball screen is an outside player.
Screened	The player receiving the ball screen is an inside player.
	The player receiving the ball screen is an outside player.
Collective Defence Type	The ball screen is performed against a man-to-man defence.
	The ball screen is performed against a zone defence.
Screening Defence Type	The defender of the dribbler player uses an over the screen defence. The defender of the dribbler player uses a under the screen defence. The defender of the dribbler player uses a switch defence against the screen. The defender of the dribbler player uses a trap defence against the screen. The defender of the dribbler player uses an ice defence against the screen.
Screening outcome	Dribbler player scored a 2- or 3-point field-goal. Pass to an open teammate. Personal foul on the screener player. Personal foul on the screened player. Missed field goal shot. Block shot. Offensive foul. Screen ends without a shot by the offensive player. Screen ends with a turnover. Screen ends with other action (out of bounds/side-line inbounds plays)
Effectiveness	Ball screen is successful.
	Ball screen is not successful.

**Table 2 ijerph-20-00059-t002:** Descriptive analysis and the χ2 intra-criteria test of the research.

Criterion	Category	n	%	χ2 Test
Team	The screening team is losing	241	46.8	χ2 = 94.303 *p* < 0.001
	The screening team is tying	70	13.6	
	The screening team is winning	204	39.6	
Scoreboard	The difference is between 0 and 3 points	271	52.6	χ2 = 219.144 *p* < 0.001
	The difference is between 4 and 6 points	110	21.4	
	The difference is between 7 and 9 points	66	12.8	
	The difference is higher than 10 points	68	13.2	
Type of offense	The action occurs in a fastbreak action	15	2.9	χ2 = 365.876 *p* < 0.001
	The action occurs in a transition action	136	26.4	
	The action occurs in a set play action	364	70.7	
Possession	0–8 s of possession remaining	80	15.5	χ2 = 132.983 *p* < 0.001
	9–16 s of possession remaining	289	56.1	
	17–24 s of possession remaining	146	28.3	
Screening Area	Top right-hand side of the court	286	55.5	χ2 = 458.056 *p* < 0.001
	Bottom right-hand side of the court	10	1.9	
	Top left-hand side of the court	208	40.4	
	Bottom left-hand side of the court	11	2.1	
Screener	Inside player	450	87.4	χ2 = 287.816 *p* < 0.001
	Outside player	65	12.6	
Screened	Outside player	511	0.8	χ2 = 499.124 *p* < 0.001
	Inside player	4	99.2	
Collective Defence Type	Man-to-man defence	509	98.8	χ2 = 491.280 *p* < 0.001
	Zone defence	6	1.2	
Screening Defence Type	The defender uses an over the screen defence	126	24.5	χ2 = 285.573 *p* < 0.001
	The defender uses an under the screen defence	90	17.5	
	The defender uses a switch defence against the screen	237	46.0	
	The defender uses a trap defence against the screen	45	8.7	
	The defender uses an ice defence against the screen	17	3.3	
Screening outcome	Dribbler player scored a 2- or 3-point field-goal	71	13.8	χ2 = 551.466 *p* < 0.001
	Pass to an open teammate	40	7.8	
	Personal foul on the screener player	13	2.5	
	Personal foul on the screened player	28	5.4	
	Missed field goal shot	134	26.0	
	Block shot	18	3.5	
	Offensive foul	7	1.4	
	Screen ends without a shot by the offensive player	168	32.6	
	Screen ends with a turnover	26	5.0	
	Screen ends with other action	10	1.9	
Effectiveness	Ball screen is successful	153	29.7	χ2 = 84.817 *p* < 0.001
	Ball screen is not successful	362	70.3	

**Table 3 ijerph-20-00059-t003:** Inter-criteria analysis between the effectiveness criterion and the rest of the study criteria.

Criterion	Category	Effectiveness			
		SCC (*n* = 153)	UNS (*n* = 362)	χ2	*p*
Team	The screening team losing	87 (36.1%)	154 (63.9%)	9.178	0.010
	The screening team is tying	15 (21.4%)	55 (78.6%)		
	The screening team is winning	51 (25%)	153 (75%)		
Scoreboard	The difference is between 0 and 3 points	73 (26.9%)	198 (73.1%)	2.357	0.502
	The difference is between 4 and 6 points	36 (32.7%)	74 (67.3%)		
	The difference is between 7 and 9 points	23 (34.8%)	43 (65.2%)		
	The difference is higher than 10 points	21 (30.9%)	47 (69.1%)		
Type of offense	The action occurs in a fastbreak action	4 (26.7%)	11 (73.3%)	8.920	0.012
	The action occurs in a transition action	27 (19.9%)	109 (80.1%)		
	The action occurs in a set play action	122 (33.5%)	242 (66.5%)		
Possession	0–8 s of possession remaining	29 (36.3%)	51 (63.7%)	5.608	0.061
	9–16 s of possession remaining	91 (31.5%)	198 (68.5%)		
	17–24 s of possession remaining	33 (22.6%)	113 (77.4%)		
Screening Area	Top right-hand side of the court	83 (29%)	203 (71%)	3.329	0.344
	Bottom right-hand side of the court	3 (30%)	7 (70%)		
	Top left-hand side of the court	61 (29.3%)	147 (70.7%)		
	Bottom left-hand side of the court	6 (54.5%)	5 (45.5%)		
Screening Defence Type	The defender uses an over the screen defence	42 (33.3%)	84 (66.7%)	14.366	0.006
	The defender uses an under the screen defence	14 (15.6%)	76 (84.4%)		
	The defender uses a switch defence against the screen	71 (30%)	166 (70%)		
	The defender uses a trap defence against the screen	20 (44.4%)	25 (55.6%)		
	The defender uses an ice defence against the screen	6 (35.3%)	11 (64.7%)		

**Table 4 ijerph-20-00059-t004:** Analysis of the success patterns of ball screens with a score in advantage as a function of score difference and time of possession.

Type of T-Pattern			N	Most Representative T-Pattern	O
-	-	-	1232	-	-
W	-	-	194	((winning_team outside_player_screened) (man-to-man_defence successful_action ))	50
W	DO3	-	45	(((winning_team 0–3_points) (set_play top_right)) ((inside_player_screener (man-to-man_defence advancement_of_screened)) successful_action)) ((winning_team (0–3_points set_play)) (inside_player_screener ((outside_player_screened man-to-man_defence) (over successful_action ))))	5 8
W	DO3	M916	15	((winning_team (0–3_points set_play)) ((possession_9´´–16´´ inside_player_screener) ((outside_player_screened man-to-man_defence) (over successful_action)))) ((winning_team ((0–3_points set_play) (possession_9´´–16´´ inside_player_screener))) ((outside_player_screened man-to-man_defence) (switch successful_action))) ((winning_team ((0–3_points set_play) (possession_9´´–16´´ inside_player_screener))) ((outside_player_screened (man-to-man_defence advancement_of_screened)) successful_action ))	7 5 5
W	D46	-	18	(((winning_team 4–6_points) (set_play top_left)) (inside_player_screener ((outside_player_screened (man-to-man_defence advancement_of_screened)) successful_action)))	4
W	D46	M08	1	((winning_team 4–6_points) ((set_play possession_0´´–8´´) (outside_player_screened (man-to-man_defence successful_action))))	4
W	D46	M916	3	((((winning_team 4–6_points) (set_play possession_9´´–16´´)) ((inside_player_screener outside_player_screened) (man-to-man_defence advancement_of_screened))) successful_action)	4
W	D79	-	9	((winning_team (7–9_points set_play)) (top_right ((inside_player_screener outside_player_screened) (man-to-man_defence successful_action))))	4
W	D79	M916	3	((winning_team (7–9_points set_play)) (possession_9´´–16´´ ((inside_player_screener outside_player_screened) (man-to-man_defence successful_action)))) ((winning_team (7–9_points set_play)) ((possession_9´´–16´´ top_left) (outside_player_screened (man-to-man_defence successful_action))))	5 4
W	D10	-	4	((winning_team (±10_points set_play)) (top_right ((inside_player_screener outside_player_screened) (man-to-man_defence successful_action))))	5
W	D10	M916	2	(((winning_team ±10 points) (set_play possession_9´´–16´´)) (top_right ((inside_player_screener outside_player_screened) (man-to-man_defence successful_action))))	4

Note. O: occurrence; W: winning; D03: the difference in the scoreboard is 0 to 3 points; D46: difference 4 to 6 points; D79: difference 7 to 6 points; D10: difference 10+ points; M916: 9–16 s of possession remaining; M08: 0–8 s of possession remaining.

**Table 5 ijerph-20-00059-t005:** Analysis of the success patterns of ball screens with the score against or tied as a function of score difference and time of possession.

Type of T-Pattern			N	Most Representative T-Pattern	O
L	-	-	350	((loosing_team inside_player_screener) ((outside_player_screened (man-to-man_defence advancement_of_screened)) successful_action))	43
L	DO3	-	66	(((loosing_team 0–3 points) (set_play inside_player_screener)) (outside_player_screened ((man-to-man_defence switch) (advancement_of screened successful_action)))) ((loosing_team (0–3_points set_play)) (top_left (inside_player_screener ((outside_player_screened man-to-man_defence) (switch successful_action)))))	8 8
L	DO3	M08	4	(((loosing_team 0–3_points) (set_play possession_0´´–8´´)) (inside_player_screener ((outside_player_screened man-to-man_defence) (over successful_action))))	5
L	DO3	M916	16	(((loosing_team 0–3 points) (set_play possession_9´´–16´´)) (top_left (inside_player_screener ((outside_player_screened man-to-man_defence) (switch successful_action))))) ((loosing_team (0–3_points set_play)) ((possession_9´´–16´´ top_left) (((inside_player_screener outside_player_screened) (man-to-man_defence pass_open_teammate)) successful_action))) ((loosing_team ((0–3_points set_play) (possession_9´´–16´´ top_right))) (inside_player_screener ((outside_player_screened (man-to-man_defence advancement_of_screened)) successful_action )))	5 4 4
L	DO3	M1724	4	((loosing_team (0–3_points transition_action)) (possession_17´´–24´´ ((inside_player_screener outside_player_screened) (man-to-man_defence successful_action))))	5
L	D46	-	68	(((loosing_team 4–6_points) (transition _action top_right)) (inside_player_screener ((outside_player_screened (man-to-man_defence advancement_of_screened)) successful_action)))	4
L	D46	M916	16	(((loosing_team 4–6_points) (set_play possession_9´´–16´´)) (inside_player_screener ((outside_player_screened (man-to-man_defemce advancement_of screened)) successful_action))) (((loosing_team 4–6_points) (possession_9´´–16´´ top_right)) (inside_player_screener ((outside_player_screened man-to-man_defence) (switch successful_action))))	6 4
L	D46	M1724	8	((loosing_team (4–6_points transition_action)) ((possession_17´´–24´´inside_player_screener) ((outside_player_screened (man-to-man_defence advancement_of_screened)) successful_action)))	5
L	D79	-	19	((loosing_team ((7–9 points set play) (inside_player_screener outside_player_screened))) ((man-to-man_defence switch) (advancement_of_screened successful_action)))	5
L	D79	M916	4	(((loosing_team 7–9_points) ((set_play possession_9´´–16´´) (inside_player_screener outside_player_screened))) (man-to-man_defence (advancement_of_screened successful_action))) (((loosing_team 7–9 points) (set_play possession_9´´–16´´)) (inside_player_screener ((outside_player_screened man-to-man_defence) (switch successful_action))))	5 5
L	D79	M1724	1	((loosing_team (7–9_points transition_action)) ((possession_17´´–24´´top_left) ((inside_player_screener outside_player_screened) (man-to-man_defence successful_action))))	4
L	D10	-	12	((loosing_team (±10_points set_play)) (top_right ((inside_player_screener outside_player_screened) (man-to-man_defence successful_action))))	6
L	D10	M916	3	((((loosing_team ±10_points) (set_play possession_9´´–16´´)) (inside_player_screener ((outside_player_screened man-to-man_defence) (switch advancement_of_screened)))) successful_action)	4
L	D10	M1724	1	((loosing_team (±10_points) ((possession_17´´–24´´outside_player_screened) (man-to-man_defence successful_action)))	5
T	-	-	14	((tying_team (0–3_points top_right)) (outside_player_screened successful_action))	8
T	DO3	-	14	((tying_team 0–3_points) (top_left ((inside_player_screener outside_player_screened) (man-to-man_defence successful_action))))	6
T	DO3	M916	8	((tying_team (0–3_points set_play)) (possession_9´´–16´´ (((inside_player_screener outside_player_screened) (man-to-man_defence pass_open_teammate)) successful_action))) (((tying_team 0–3_points) (set_play possession_9´´–16´´)) (top_right ((inside_player_screener outside_player_screened) (man-to-man_defence successful_action))))	5 5

Note. O: occurrence; L: losing; T: tying; D03: the difference in the scoreboard is 0 to 3 points; D46: difference 4 to 6 points; D79: difference 7 to 6 points; D10: difference 10+ points; M916: 9–16 s of possession remaining; M08: 0–8 s of possession remaining.

## Data Availability

Not applicable.

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
