# Peer review of "Ball Screens in the Men’s 2019 Basketball World Cup"

_ijerph, 2022, doi:10.3390/ijerph20010059_

Round 1

Reviewer 1 Report

Observational study conducted through video analysis of matches of the 2019 men's basketball world cup.

In the Introduction section, the authors state the frequency with which the action they intend to investigate is applied, showing data from previous studies, although they do not go into the need and relevance of the study in depth.

The methodology applied is correct, although the results should be explained in greater detail to facilitate the reader's understanding. For example, the use of abbreviations is abused. The tables should have table captions with legends about these abbreviations.  It is recommended that table 4 be broken down into at least two smaller tables to facilitate analysis.

In the Discussion section, the fact that similar studies are not discussed makes this section poor as presented. Authors are advised not to simply state their results by comparing them with previous results, but to obtain solid arguments that connect well with the need for this study, as we suggest you present in the Introduction.

The key words should be revised, not being found in the text.

Reviewer 2 Report

Overall Comments

This study aimed to analyze the effectiveness and successful patterns in ball screens in the men’s 2019 Basketball World Cup. The paper is well-written and provides important data in basketball research, particularly in such crucial offensive play as the ball screen. I recommend that the paper should be accepted after some minor revisions.

Introduction

The Introduction is well-written and provides an adequate background on the topic.

Materials and Methods

Design: The last sentence of this section (line 87) seems incomplete. Please confirm.

Sample: The definition of an unsuccessful screen includes a missed 2- or 3-point shot. However, the purpose of the screen is to provide better opportunities to shoot. If a player misses an open shot following a screen, that should not interfere with the screen's movement. I understand the authors’ point of view, but I recommend including this idea in the discussion section, probably as a limitation. 

Results

I think that the “statistical analysis” section should be deleted since it presents the research's overall results (line 165). Also, the first sentence seems incomplete (line 166), and the authors should use the number of the Table when referring to it in the text.

Tables 2, 3, and 4: Please provide the abbreviations subtitles for a better experience for the reader. 

Discussion

The Discussion is well-established. Congratulations to the authors for their work.

Round 2

Reviewer 1 Report

We consider that the modifications to the article have improved the quality of the manuscript, although the appearance of the yellow colour (which is not mentioned in his response letter to reviewers) leaves us uneasy, as we do not know whether these issues were finally eliminated or are part of this version of the manuscript. 

Author Response

Dear Reviewer,

Thank you very much for your comments and for the new revision. 
In response to your comment, we would like to point out the following:

The text underlined in yellow is final text of the document. We underlined it to indicate that it was a change from the original document.

In the current version we have removed the yellow underlining and present the final document, unless you feel we need to change anything else.

Again, thank you very much for your work.